# Postnatal Cytoarchitecture and Neurochemical Hippocampal Dysfunction in Down Syndrome

**DOI:** 10.3390/jcm10153414

**Published:** 2021-07-31

**Authors:** David G. Moreno, Emma C. Utagawa, Nicoleta C. Arva, Kristian T. Schafernak, Elliott J. Mufson, Sylvia E. Perez

**Affiliations:** 1Department of Neurobiology, Barrow Neurological Institute, Phoenix, AZ 85013, USA; David.Moreno@Barrowneuro.org (D.G.M.); Emma.Utagawa@Barrowneuro.org (E.C.U.); Elliott.Mufson@Barrowneuro.org (E.J.M.); 2Department of Pathology and Laboratory Medicine, Ann & Robert H. Lurie Children’s Hospital of Chicago, Chicago, IL 60611, USA; NArva@luriechildrens.org; 3Department of Pathology and Laboratory Medicine, Phoenix Children’s Hospital, Phoenix, AZ 85016, USA; kschafernak@phoenixchildrens.com

**Keywords:** down syndrome, hippocampus, postnatal development

## Abstract

Although the prenatal hippocampus displays deficits in cellular proliferation/migration and volume, which are later associated with memory deficits, little is known about the effects of trisomy 21 on postnatal hippocampal cellular development in Down syndrome (DS). We examined postnatal hippocampal neuronal profiles from autopsies of DS and neurotypical (NTD) neonates born at 38-weeks’-gestation up to children 3 years of age using antibodies against non-phosphorylated (SMI-32) and phosphorylated (SMI-34) neurofilament, calbindin D-_28k_ (Calb), calretinin (Calr), parvalbumin (Parv), doublecortin (DCX) and Ki-67, as well as amyloid precursor protein (APP), amyloid beta (Aβ) and phosphorylated tau (p-tau). Although the distribution of SMI-32-immunoreactive (-ir) hippocampal neurons was similar at all ages in both groups, pyramidal cell apical and basal dendrites were intensely stained in NTD cases. A greater reduction in the number of DCX-ir cells was observed in the hippocampal granule cell layer in DS. Although the distribution of Calb-ir neurons was similar between the youngest and oldest NTD and DS cases, Parv-ir was not detected. Conversely, Calr-ir cells and fibers were observed at all ages in DS, while NTD cases displayed mainly Calr-ir fibers. Hippocampal APP/Aβ-ir diffuse-like plaques were seen in DS and NTD. By contrast, no Aβ_1–42_ or p-tau profiles were observed. These findings suggest that deficits in hippocampal neurogenesis and pyramidal cell maturation and increased Calr immunoreactivity during early postnatal life contribute to cognitive impairment in DS.

## 1. Introduction

Down syndrome (DS), a genetic disorder caused by trisomy of chromosome 21 (Ch21), is characterized by substantial developmental delay and intellectual disability during childhood and adulthood [1,2,3,4], which further deteriorate with the age-related onset of amyloid beta (Aβ) plaque and tau bearing neurofibrillary tangle (NFT) pathology similar to that observed in Alzheimer’s disease [5,6,7,8]. Children with DS perform worse on short-term memory (STM) and long-term memory (LTM) tasks compared to neurotypically developing (NTD) children, suggesting that structures underlying memory function, such as the hippocampus, are compromised in DS at early developmental stages [9,10,11,12,13,14]. Gestational weeks 10, 15–19, 25, and 34 are crucial periods for the normal development of the human hippocampus which grows steadily from birth to the age of 14 years [15,16]. Studies have found early developmental decreases in hippocampal volume and neuronal density during these critical gestation weeks in individuals with DS [17,18,19]. These decreases have been attributed to impaired neurogenesis during fetal gestation weeks 17–23 [20] due to the overexpression of amyloid precursor protein (APP) and its metabolite Aβ_1–42,_ resulting from trisomy 21 as early as 21 gestational weeks [21]. Giacomini and colleagues [22] demonstrated that APP gamma-secretase enzyme inhibition restored hippocampal neurogenesis, hippocampal granule cell number and synaptic development in postnatal Ts65Dn mice, suggesting that APP plays a critical role in hippocampal neurogenesis. Moreover, APP has also been implicated in neuronal differentiation and synaptogenesis during the brain development in DS [23,24]. In this regard, reduced hippocampal GABA_A_ subunit alpha-3 was associated with increased APP/Aβ_1–42_ levels in the DS fetus and neuronal differentiation in a cellular model of DS [25]. Phosphorylation of human fetal tau (N03R), which is found in distal portions of growing axons and is downregulated after axons reach their target sites [26], is disturbed early in DS hippocampal development [27]. Moreover, a general delay in myelin formation and a decrease in myelinated axons within the hilus of the dentate gyrus (DG) was observed from the fetal period to adulthood in DS, resulting in amelioration of axonal neurotransmission [28].

Together, these observations suggest that, during fetal development, alterations in hippocampal development occur, affecting the structure and function of this component of the medial temporal lobe memory circuit in DS. However, the effects of trisomy 21 on hippocampal neuronal differentiation during early postnatal development in DS remain underinvestigated. In this study, we examined the neurochemical spatiotemporal development of neuronal profiles containing the intermediate cytoskeletal non-phosphorylated and phosphorylated high-molecular-weight neurofilament proteins; the GABAergic interneuronal calcium binding proteins (CBP) Calbindin D-_28k_ (Calb), Calretinin (Calr), and Parvalbumin (Parv); the neuronal migration marker microtubule-associated protein doublecortin (DCX) and the cellular proliferation marker Ki-67 in the hippocampal Cornu Ammonis (CA) subfields CA1-CA3, DG and subiculum (Sub) from neonates born at 38-weeks’-gestation up to children 3 years of age in DS and NTD tissue collected postmortem. In addition, we also examined APP/Aβ and phosphorylated tau (p-tau) reactivity at these ages.

## 2. Experimental Section

### 2.1. Subjects and Tissue Samples

This study used postmortem tissue from a total of 17 liveborn subjects, which were characterized as neurotypical (*n* = 9) or karyotyped as DS (*n* = 8) ranging from 38 weeks of gestation to 3 years of age (196 weeks). DS cases were obtained from either Phoenix Children’s Hospital (PCH) (*n* = 2) or the Ann & Robert H. Lurie Children’s Hospital of Chicago (LCH) (*n* = 6), while NTD cases were accrued exclusively from PCH (*n* = 9). Tissue was obtained and processed according to IRB guidelines meeting the exemption criteria 45 CFR 46.101 (b) and managed under the Barrow Neurological Institute procedures. Table 1 lists case demographics: sex, body weight, body height (crown to heel), brain weight, postmortem interval (PMI), gestational age at birth and death, postnatal life between birth and death, tissue source, and cause of death or comorbidity. Please note that the ages reported here combine the number of gestational weeks plus the number of weeks of postnatal life. Forty weeks was considered a full-term pregnancy [29]. Brain tissue was fixed in 10% neutral buffered formalin and embedded in paraffin. Blocks containing the posterior hippocampus and subiculum were sectioned at 8 µm with the exception of cases 5 and 7 which were cut at 4 µm-thickness, placed on charged slides, and stored at room temperature (RT) until processing.

### 2.2. Immunohistochemistry

Two hippocampal sections from each case were single-immunostained using antibodies directed against APP/Aβ (6E10), Aβ_1–42_, phosphorylated CP-13, PHF-1 and AT8 tau, phosphorylated SMI-34 and non-phosphorylated SMI-32 neurofilaments (Nfl), Ki-67, DCX, and the CBP Calb, Parv, and Calr (see Table 2 for details). All sections were deparaffinized in xylene, hydrated in a series of degraded concentrations of ethanol (100, 95, 70, and 50%) and pretreated with either heated citric acid (pH 6) for 10 min in a microwave, or with 88% formic acid for 20 min for antigen retrieval of APP/Aβ and Aβ_1–42_ proteins. After several washes in Tris-buffered saline (TBS), sections were incubated in sodium metaperiodate (0.01 M) solution for 20 min to inactivate endogenous peroxidase activity, then rinsed three times in TBS, and placed in blocking solution containing 0.25% Triton X-100/3% goat serum (GS) for one hour (h) at RT and then incubated with primary antibodies in a solution of TBS 0.25% Triton X-100/1% GS overnight at RT or two nights at 4 °C (see Table 2). After several 1% GS TBS washes, tissue was incubated with a goat anti-mouse or anti-rabbit biotinylated secondary antibody (1:200; 1 h) (Vector Labs, Burlingame, CA) as appropriate, incubated in Vectastain ABC kit (1 h) (Vector Labs) and developed in acetate-imidazole buffer containing 0.05% 3,3′-diaminobenzidine tetra hydrochloride (Thermofisher Scientific, Waltham, MA, USA) and 0.005% H_2_O_2_, resulting in a brown reaction product [8,30]. Select sections were counterstained with Mayer’s hematoxylin for cytoarchitectural cellular analysis. All slides were dehydrated, cleared in xylenes and coverslipped using DPX (Electron Microscopy Sciences, Hatfield, PA, USA). To control for batch-to-batch variation in immunostaining procedures, sections from each case were processed at the same time for each antibody. Frontal cortex paraffin embedded tissue from an 82-year-old female with neuropathologically confirmed AD (Braak NFT stage VI), hippocampal sections from a DS 20-weeks’-gestation fetus, NTD 22-weeks’-gestation fetus, and anaplastic astrocytoma tumor (8 year-old female) were used as positive controls for tau [26], Aβ and Ki-67 antibodies staining, respectively. Primary antibody omission resulted in absence of immunostaining for each antibody. To test the specificity of 6E10 immunostaining, this antibody was preabsorbed against purified human Aβ_1–17_ (AnaSpec Inc., Fremont, CA, USA) at a concentration of 50–500 µg/μL or Aβ_1–42_ (AnaSpec Inc.) at concentration of 50μg/µL overnight, followed by immunolabeling of postnatal hippocampal tissue with the above protocol. In addition, we performed control immunohistochemical antibody experiments including omission of the primary, secondary and both the primary + secondary antibodies. To further confirm the 6E10 immunoreactivity, we stained AD frontal cortex tissue from a 82-year-old female, hippocampal tissue from a NTD 12-year old, as well as frontal cortex and cerebellar cortex sections from a 28 and 174 week-old NTD and from a 31 and 196 week-old DS case (Appendix A). Reacted sections were photographed using a Nikon Eclipse 80i, and contrast and brightness were adjusted using Adobe Photoshop (Adobe, San Jose, CA, USA). The nomenclature of the postnatal hippocampus was based on the developmental studies of Insausti and colleagues [16,31].

### 2.3. Immunofluorescence

All sections were deparaffinized with xylene, hydrated, and pretreated with heated citric acid (pH 6) in a microwave for 10 min. Sections were then incubated with a monoclonal against DCX (1:25; Santa Cruz Biotechnology, Dallas, TX, USA) in a solution containing TBS 0.25% Triton X-100/1% donkey serum overnight at RT. After several washes in TBS containing 1% donkey serum, sections were incubated with a donkey anti-mouse Cy2 conjugated affinity secondary antibody (1:200; 1 h) (Jackson ImmunoResearch Laboratories, Inc., Chester County, PA, USA). Sections were then incubated with polyclonal anti-Calb (1:75; Swant, Marly, Switzerland) antibody in a TBS 0.25% Triton X-100/1% donkey serum solution overnight at RT. The next day, sections were incubated in a solution that contained donkey anti-rabbit Cy5 conjugated affinity secondary antibody (1:200; 1 h) (Jackson ImmunoResearch Laboratories, Inc., Chester County, PA, USA). Slides were then coverslipped using Invitrogen ProLong Glass Antifade Mountant (Invitrogen, Carlsbad, CA, USA). Immunofluorescence was photographed using an Echo Revolve fluorescence microscope (San Diego, CA, USA).

### 2.4. Histochemistry

Hematoxylin/eosin (H&E) and thionin stains were used to define the cytoarchitecture of the hippocampus. Thionin was used to perform neuronal counts due to its greater affinity to stain neuronal Nissl substance. One section per sections were deparaffinized with xylene, hydrated, and pretreated with heated citric acid (pH 6) in a microwave for 10 min. Sections were then incubated with a monoclonal against DCX (1:25; Santa Cruz Biotechnology, Dallas, TX, USA) in a solution containing TBS 0.25% Triton X-100/1% donkey serum overnight at RT. After several washes in TBS containing 1% donkey serum, sections were incubated with a donkey anti-mouse Cy2 conjugated affinity secondary antibody (1:200; 1 h) (Jackson ImmunoResearch Laboratories, Inc., Chester County, PA, USA). Sections were then incubated with polyclonal anti-Calb (1:75; Swant, Marly, Switzerland) antibody in a TBS 0.25% Triton X-100/1% donkey serum solution overnight at RT. The next day, sections were incubated in case was deparaffinized, hydrated in descending gradients of alcohol followed by distilled water, and placed in a hematoxylin solution for 2 min, washed with running tap water for 5 min, placed into eosin for 2 min, dipped in distilled water, dehydrated and coverslipped with DPX (Electron Microscopy Sciences, Hatfield, PA, USA). Two other deparaffinized slides from each case were incubated in 100% ethanol for 3 min, then placed into a 50% chloroform/50% ethanol solution for 15 min, rehydrated in alcohol, soaked in a 0.5% thionin (pH 4.3) solution for 7 min, rinsed with distilled water, dehydrated in increased concentrations of alcohol, cleared in xylenes, and coverslipped with DPX. To control for batch-to-batch variation in H&E and thionin staining, sections from each case were processed at the same time blinded to the experimental group.

### 2.5. Cell Quantitation

Counts of thionin, SMI-32, Calb, Calr and DCX positive cells and 6E10 plaques were performed in three randomly sampled areas of the DG, CA2/3, and CA1 subfields of the hippocampus at 400× magnification from two slides using a Nikon Eclipse 80i, and the average number of neurons per region per case was calculated. Due to a reduced number of cases containing the Sub we were not able to perform statistical analysis. Thionin cell counts were used to normalize immunohistochemical neuronal counts and were presented as percentages. An investigator blinded to demographics performed counts.

### 2.6. Statistical Analysis

Normalized and non-normalized neuron counts and case demographics were compared between groups using a non-parametric Mann—Whitney sum rank test, Fisher exact test, and Kruskal—Wallis test followed by Dunn’s post hoc test for multiple comparisons. In addition, we performed Friedman repeated measures analysis of variance on ranks followed by a post hoc Tukey test (Sigma Plot 14.0, Systat Software, San Jose, CA, USA). Correlations were performed with a Spearman rank sum analysis. Significance (*p* value) was set at less than 0.05 (two-tailed). Cell count data and correlations were graphically represented as box-plots and linear regressions, respectively using Sigma Plot 14.0 Software.

## 3. Results

### 3.1. Case Demographics

There were no significant differences for age, brain weight, body weight and height (crown to heel) between NTD and DS cases (Mann—Whitney rank; *p* > 0.05, Table 3). Average age was 64.87 (range, 50–80) weeks for NTD and 72.80 (range, 54–91) weeks for DS; NTD brain weight was 653.08 g (range, 555–750) and 544.75 g (range, 443–646) for DS; NTD body weight was 6.38 kg (range, 2.5–13) and 6.036 kg (range, 2.5–14) for DS cases; NTD height was 61.66 cm (range, 39–90) and 61.45 cm (range, 48–97) for DS cases. No significant differences were found for sex (Fisher exact test; *p* > 0.05) or PMI (Mann—Whitney rank; *p* > 0.05) between groups.

### 3.2. Postnatal Hippocampal Complex Cytoarchitecture

Thionin and H&E staining revealed greater neuronal organization and development in the hippocampus, DG and Sub in the NTD compared to DS cases (Figure 1 and Figure 2). Since thionin histochemistry provided greater morphological cellular clarity, it was used to describe postnatal hippocampal cytoarchitecture and quantitation. We found that neurons in the hilus, CA1–3 subfields and Sub were more intensely thionin positive in NTD (Figure 2Q–T,V-Y,Z1–Z4) than in DS (Figure 2B–E,G–J,L–O). Although we observed DG, CA2/3 and Sub pyramidal-shaped cells at the youngest ages (DS, 40 weeks; NTD, 38 weeks) in both groups, apical and basal dendrites were more prominent in NTD cases (Figure 2A,C–E,P,Q–T). At these ages, DG hilar and CA1 neurons appeared disorganized and displayed lightly thionin-stained processes (Figure 2B,D,Q,S). By 45 weeks, DS hilar neurons were larger than at 40 weeks, but this change was not seen for CA1 pyramidal neurons (Figure 2G,I). In the oldest cases (DS 196 weeks; NTD 174 weeks), hilar, CA1–3, and Sub thionin-stained neurons appeared larger than in the youngest cases in both groups (Figure 2L–O,Z1–Z4). However, CA1 thionin-stained pyramidal neurons (Figure 2N) were smaller compared to CA3 and Sub pyramidal cells in DS (Figure 2L,O) and NTD CA1 neurons (Figure 2Z3).

Quantitation of thionin-stained DG and hippocampal CA cells revealed comparable numbers in both NTD and DS. A within group analysis found a significantly higher number of DG compared to CA1 neurons in DS (Friedman repeated measures, *p* = 0.003) and NTD (Friedman repeated measures, *p* = 0.004) cases, while the DG contained higher numbers of cells than the CA2/3 pyramidal cell layer (PL) in NTD (Figure 3; Friedman repeated measures, *p* = 0.043).

### 3.3. Hippocampal APP/Aβ, Aβ_1–42_ and Phosphorylated Tau Immunoreactivity

Bright-field microscopy revealed diffuse bursts/deposits positive for APP/Aβ (6E10), but not Aβ_1–42_, scattered throughout the DG and hippocampal CA subfields in both groups at all ages (Figure 4A,B,D,F,G). Although qualitatively hippocampal APP/Aβ immunoreactive (-ir) deposits appeared more abundant in DS (Figure 4A) than in NTD (Figure 4D), quantitation revealed no significant differences between groups (*p* > 0.05). Preabsorption of the 6E10 antibody with Aβ_1–17_ and Aβ_1–42_ peptides and omission of the primary 6E10, secondary and 6E10 + secondary antibodies resulted in the absence of APP/Aβ immunostaining in both NTD and DS postnatal hippocampus (Figure 4C,E). We also found similar hippocampal APP/Aβ immunostaining in a 12-year-old NTD case as well as in the postnatal frontal and cerebellar cortex in both DS and NTD cases that was not seen in control immunohistochemical experiments (Appendix A).

The hippocampus was immunonegative for p-tau (CP-13, PHF-1 and AT8) at each postnatal time point in both groups examined. However, CP-13 positive fibers were seen in the fimbria, alveus and molecular layer (ML) in a 20-week DS and 22-week gestation NTD case (Figure 5A,B), similar to prior findings [26].

### 3.4. Neurofilament Positive Profiles in the Postnatal Hippocampal Complex

SMI-32 and SMI-34 antibodies were used to reveal the presence of non-phosphorylated and phosphorylated heavy molecular weight Nfl, a neuronal maturation marker, in the postnatal hippocampus. SMI-32 preferentially labels non-phosphorylated Nfl in the soma and dendrites, while SMI-34 recognizes phosphorylated intermediate Nfl of high-molecular-weight mainly in axons. Bright-field examination showed strong SMI-32-ir pyramidal cells in the Sub and large multipolar polymorphic cells in the dentate hilus, while pale staining was observed in pyramidal cells in CA2/3 in DS and NTD cases at all ages (Figure 6). In both groups, hippocampal SMI-32-ir was stronger in the oldest (196 weeks) compared to the younger (45 weeks) cases (Figure 6A,K). SMI-32-ir pyramidal cells in the Sub and CA2 region and hilar interneurons (Figure 6D,H) were observed in the youngest cases in both groups (DS 40 weeks; NTD 38 weeks), while CA1 pyramidal cells were only lightly stained at these ages (Figure 6E,I). Granule cells in the DG showed weaker SMI-32 immunostaining compared to Sub pyramidal cells in DS and NTD in all cases (Figure 6C,F,G,J). DG SMI-32-ir cells exhibited stronger immunostaining in NTD than DS (Figure 6C,G). At 44 weeks, pyramidal neurons in CA1 and Sub and DG granule cells displayed similar morphology to that observed in the oldest NTD cases showing a large apical and one or more basal dendrites compared to DS, which displayed less-developed processes (Figure 6C,F,G,J).

Quantitation revealed no significant differences in SMI-32 positive DG, CA2/3 and CA1 cells between groups (Mann—Whitney rank; *p* > 0.05). On the other hand, a within group analysis revealed that the numbers of DG positive cells were significantly higher compared to CA1 cell values in NTD (Figure 6O; Friedman repeated measures; *p* = 0.004), which was not seen in DS (Figure 6O; Friedman repeated measures, *p* > 0.05). Moreover, the percentage of DG SMI-32-ir cells normalized to thionin counts were significantly higher in NTD compared DS, while within groups, the percentage of CA1 SMI-32-ir cells were greater than DG granular cells in DS, but not in the NTD group (Friedman repeated measures, *p* = 0.036)

Tissue from the youngest DS cases (40 to 53 weeks) stained for SMI-34 exhibited very few fibers and cells in the hilus and CA3 subfield (Appendix A). At NTD 38 weeks, only SMI-34-ir fibers were observed coursing in the alveus/stratum oriens (Appendix A). By 44 weeks, SMI-34-ir fibers were observed in the DG ML as well as in the alveus/stratum oriens, which became stronger from 48 to 60 weeks in the NTD group. At these ages SMI-34-ir fibers were also observed in the stratum radians/lacunosum moleculare in NTD. At DS 61 weeks, there were scarce fibers in the DG hilus, however, at 83 weeks many more SMI-34-ir fibers were observed in the hilus, alveus/stratum oriens and radians/lacunosum moleculare. At the oldest ages (NTD: 174 weeks, DS: 196 weeks) SMI-34-ir fibers were seen coursing in the alveus/oriens (Appendix A) and lacunosum-moleculare, and a few fibers appeared in stratum radians and hilus in both groups. Scarce number of SMI-34-ir cells were observed in close proximity to the alveus/oriens positive fibers in DS (Appendix A).

### 3.5. Postnatal Hippocampal Cellular Proliferation and Neurogenesis

Here, we used an antibody against Ki-67, a nuclear protein expressed only during cell division, to examine postnatal hippocampal proliferation, while neurogenesis was evaluated with DCX, a microtubule-associated protein expressed in neuroblasts/migrating neurons [32]. Unlike Ki-67, which was not immunodetected in the postnatal hippocampus at any age in either group, DCX was observed mainly in the DG and CA subfields in younger DS and NTD cases (38 to 53 weeks; Figure 7A,B). Strong DCX immunostaining was found in the granule cell layer (GL) closest to the hilus throughout the DG, which was weaker in CA1, CA2/3 and Sub pyramidal neurons in both groups at 38–83 weeks (Figure 7A,B). These DG fusiform-shaped DCX-ir cells with a process coursing through the DG GL towards the ML displayed a smaller nucleus compared to immunonegative cells (Figure 7C,D). At the light microscopic level, we observed fewer DG DCX-ir cells with increasing age in both groups (Figure 7E,F, and Appendix A). After 61 weeks of age less DCX positive cells were observed in DS than in NTD (Figure 7E,F and Appendix A) and none were visualized at either week 196 in DS or at 174 in NTD (Figure 7G,H and Appendix A).

Quantitative counts did not show a significant difference between the number of DCX-ir cells in DG, CA1 and CA2/3 fields between the DS and NTD groups (Figure 8; Mann—Whitney rank, *p* > 0.05). Counts also revealed that DCX positive cells in the DG were significantly greater compared to CA1 in the NTD group (Figure 8; Friedman repeated measures; *p* = 0.031), which was not seen in DS (Figure 8; Friedman repeated measures, *p* > 0.05). We did not observe a difference in the percentage of DCX positive cells in DG, CA1 and CA2/3 fields between or within groups.

### 3.6. Postnatal Hippocampal Calcium Binding Protein Immunoreactivity

Hippocampal CBP profiles were visualized using antibodies against Calb, Parv, and Calr. Granule cells in the DG exhibited strong Calb-ir in both groups at all ages but neuronal morphology was better defined in the youngest compared to the oldest cases in both groups (Figure 9A–D). In the youngest cases, CA2 pyramidal neurons displayed lighter reactivity compared to the oldest cases in both groups (Figure 9I–L). On the contrary, CA1 Calb-ir pyramidal neurons were less well-stained in the oldest compared to the youngest (Figure 9M–P) cases. In addition, small non-pyramidal Calb cells in CA subfields were observed at all ages in each cohort (Figure 9M–P). Calb-ir fibers surrounding the pyramidal cells were seen within the CA3 subfield at all ages in both groups (Figure 9E–H). In the oldest cases, fine beaded Calb-ir fibers were observed in CA1 (Figure 9N,P) in DS and NTD cases. Calb-ir cell numbers were not significantly different in any region examined between groups, (Figure 9Q; Mann—Whitney rank; *p* > 0.05). Conversely, DG Calb positive granule cell numbers were significantly higher compared to CA1 (Figure 9Q; Friedman repeated measures *p* = 0.033) and CA2 Calb-ir counts within the NTD group, (Figure 9Q; Friedman repeated measures; *p* = 0.001), but not in DS (Figure 9Q; Friedman repeated measures, *p* > 0.05). Normalized data revealed no differences in the percentage of Calb-positive cells between or within groups (*p* > 0.05).

Parv immunostaining was not detected in any region of the postnatal hippocampus at any age in either group. Conversely, Calr-ir cells and fibers were observed in DS (Figure 10) but only Calr-ir fibers were seen in NTD cases. In the DG of all DS cases, we found small fusiform Calr-positive cells in the GL, large multipolar cells in the hilus, (Figure 10A–E) and fibers adjacent to the GL. Elongated Calr-ir cells and a band of fibers were observed between the DG ML and stratum lacunosum-moleculare (Figure 10F–H) as well as between the hippocampal stratum radians and stratum lacunosum-moleculare (Figure 10I) and in the stratum oriens (Figure 10J). Additionally, a few scattered small Calr-positive cells and fibers were observed in the CA1 PL (Figure 10E) and stratum oriens (Figure 10J). At 83 weeks, fewer Calr-positive cells were seen throughout the DG and CA subfields compared to prominent fiber staining (Figure 10K–Q) in the hilus (Figure 10T) and DG ML (Figure 10R). In the oldest DS case (196 weeks), Calr-ir cell staining was stronger in the polymorphic layer, which displayed multiple processes compared to younger cases (Figure 10R–V). At this age, Calr-ir cells and fibers were seen in the CA1 PL (Figure 10W) as well as between the stratum lacunosum-moleculare (Figure 10X,Y) and stratum radians (Figure 10Z,Z1). Comparable to DS, the youngest NTD cases displayed Calr-ir fibers in the DG ML and hilus, between the DG ML and stratum lacunosum-moleculare, and between the lacunosum-moleculare and radians and oriens strata (Figure 10Z2). A few scattered small Calr-ir cells were found in the PL of CA subfields. Calr-ir fibers were more abundant in the oldest cases (Figure 10Z3). Quantitation revealed no significant differences in Calr-ir cell number between hippocampal areas in DS (Figure 10Z4, Friedman repeated measures; *p* > 0.05).

### 3.7. Colocalization of DCX and Calb in the DG

Double immunofluorescence revealed Calb and DCX-containing DG granule cells mainly at the youngest ages in both groups (Figure 11). Calb/DCX-dual labeled cells displayed a small fusiform undifferentiated morphology as in the single DG DCX-positive cells compared to larger rounded single Calb-positive granule cells (Figure 11A,D). Although dual Calb/DCX and single DCX-ir cells in the DG GL decreased with age in both groups (Figure 11A–F), at the oldest ages only Calb-ir cells were found in the GL in NTD and DS (Figure 11C,F) cases.

### 3.8. Correlations

In NTD, DG DCX-ir cell counts displayed a strong positive correlation with CA1 (Appendix A; Spearman rank, *r* = 0.96 *p* = 0.0000002) and CA2 Calb (Appendix A; Spearman rank, *r* = 0.85 *p* = 0.00609) positive cell counts. CA1 Calb-ir pyramidal cell counts exhibited a strong positive correlation with CA2 Calb-ir neurons (Appendix A; Spearman rank, *r* = 0.88 *p* = 0.0000002). Within DS, CA1 DCX-ir counts correlated with DG SMI-32 (Appendix A; Spearman rank, *r* = 1.00 *p* = 0.0027). Moreover, DG DCX-ir cell numbers showed a strong negative correlation with brain weight in DS (Figure 12A; Spearman rank, *r* = –0.96 *p* = 0.0000002); but weaker in NTD (Figure 12A; Spearman rank, *r* = –0.83 *p* = 0.005). CA2/3 thionin cell counts showed a negative association with brain weight (Spearman rank, *r* = −0.9 *p* = 0.0000002) and age (Spearman rank, *r* = −0.85 *p* = 0.006) in NTD, but not in DS. Negative correlations were observed between DG DCX-ir counts and age in NTD (Figure 12B; Spearman rank *r* = −0.83 *p* = 0.005) and DS (Figure 12B; Spearman rank, *r* = –0.78 *p* = 0.025). CA2/3 DCX cell counts correlated negatively with age in DS (Figure 12C; Spearman rank, *r* = −1.00 *p* = 0.0027), but not in NTD. By contrast, CA1 Calb-ir pyramidal cell counts showed a negative correlation with brain weight (Figure 12D; Spearman rank, *r* = −0.92 *p* = 0.0000002) and age (Figure 12E; Spearman rank, *r* = −0.85 *p* = 0.0017) in NTD.

Body weight correlated negatively with DCX DG (Figure 13A; Spearman rank, *r* = −0.91 *p* = 0.0000002), CA2/3 thionin (Spearman rank, *r* = −0.86 *p* = 0.006) and Calb CA1 (Figure 13B; Spearman rank, *r* = −0.86 *p* = 0.0018) counts in NTD. In DS, correlations between body weight and DG DCX (Figure 13A; Spearman rank, *r* = −0.86 *p* = 0.0061), DG thionin (Spearman rank, *r* = −0.81 *p* = 0.0096), DG SMI-32 (Figure 13C; Spearman rank, *r* = −0.86 *p* = 0.0061), CA2/3 DCX (Figure 13D; Spearman rank, *r* = −1.0 *p* = 0.0028) and CA1 thionin (Spearman rank, *r* = −0.85 *p* = 0.006) cells were less significant. Interestingly, height had a negative correlation with DG SMI-32 (Appendix A; Spearman rank, *r* = −0.86 *p* = 0.0061), DG DCX (Appendix A; Spearman rank, *r* = −0.85 *p* = 0.0061), and CA2/3 DCX (Appendix A; Spearman rank, *r* = −1.0 *p* = 0.0028) in DS, and with Calb-ir CA1 (Appendix A; Spearman rank, *r* = −0.83 *p* = 0.0053) counts in NTD. Finally, correlations between age and brain weight were stronger in the NTD compared to DS (Figure 14; Spearman rank, NTD *r* = 0.98 *p* = 0.0000002, DS *r* = 0.85 *p* = 0.0017).

## 4. Discussion

Prenatal and postnatal development of the brain comprises a series of complex coordinated events including cell proliferation and migration, neuronal enlargement, axonal and dendritic growth, synaptic formation, glial proliferation and myelination. Disturbances in these processes underlie structure-function dysfunction including cognition during development. Prior neuropathological studies revealed abnormalities in the developing DS brain including impaired neuronal migration, hypocellularity in the hippocampal DG, and reduced cerebellar volume during the late prenatal period [2]. While many of these processes have been reported during prenatal stages, there is a lack of information on brain development during early postnatal stages, particularly in the hippocampal formation, a limbic structure involved in cognition, learning and memory [12,13,14]. The hippocampal complex is composed of several regions including the DG, CA subfields, Sub, and the fimbria and alveus white matter tracts [31]. To understand the effect of trisomy 21 on postnatal neurodevelopment of the hippocampus, we examined tissue obtained postmortem at different postnatal ages from DS and NTD cases.

### 4.1. Postnatal Hippocampal Neurogenesis in DS

Defects in cell proliferation and migration during brain development result in neurological disorders. Although the nuclear protein Ki-67, a marker of dividing cells [33,34], appears in human DG cells and germinal zones of the hippocampus proper and the germinal matrix of the inferior horn of the lateral ventricles during gestational weeks 17–21, DS fetuses display a decreased number of these cells compared to NTD cases [35]. In addition, DG Ki-67-ir cells decrease drastically during and following the first year of life, with only a few dividing neurons found in the hilus and DG granule cells between years 1 to 35 [36]. In the present study, Ki-67-ir cells were not seen in the DG and hippocampus at the early postnatal stages in both the DS and NTD cases examined here. The lack of Ki-67 cells in the postnatal DG could be due to methodological differences but likely indicate postnatal curtailment of cell division [37].

The assembly of neuronal circuits relies on neuronal migration occurring in the appropriate spatio-temporal pattern. In the present study, DCX was used to evaluate postnatal neurogenesis in the hippocampus. DCX is a microtubule-associated protein expressed during embryonic and postnatal development in migrating neurons in the central and peripheral nervous system [38,39], which is crucial for the formation of neural circuits [40]. Previous studies found that mice with a mutation in the gene encoding DCX displayed disruption of hippocampal cytoarchitecture and hippocampus-based learning [41]. In the present study, DCX-ir cells were notoriously observed in the DG compared to other regions of the hippocampus. A recent study demonstrated that the numbers of DCX-positive cells declined sharply in the first year of life in the human DG of non-DS subjects [36]. Although there was an age-related decline in DG DCX immunostaining in the hippocampus in both groups, less DG DCX positive cells were seen after 61 weeks in DS. The differential expression of DCX positive cells in the GL between groups suggests a deficit in neuronal migration/neurogenesis during postnatal development in DS [35]. Interestingly, we found a stronger negative correlation between DG DCX-ir cell counts and brain weight in DS compared to NTD. We also demonstrated a closer association between brain weight and age in NTD than in DS, where DS brain growth rate was lower than the NTD cases (see Appendix A). These observations indicate linkage between hippocampal neurogenesis and brain/body growth during postnatal ages.

### 4.2. Postnatal Hippocampal Cytoarchitecture in DS

In this study, thionin histochemistry was used to characterize the cytoarchitecture of the CA fields, DG and Sub in DS and NTD [42]. Tissue stained for thionin revealed a well-defined neuronal organization and development in the DG, CA fields and Sub in all NTD cases examined. At 38 weeks, CA subfields and Sub pyramidal neurons, as well as hilar interneurons, displayed stronger cytoplasmic and neuronal processes, which were only lightly stained in the younger DS cases. The difference in staining may reflect less well-developed Nissl substance of the rough endoplasm reticulum (RER) in DS hippocampal neurons. Since the RER plays a role in protein synthesis, lipid synthesis, calcium homeostasis, neuronal function and activity [43], it is likely that postnatal hippocampal neurons display reduced neuronal activity in DS. Interestingly, we also observed that thionin-stained pyramidal neurons in the CA1 field are smaller compared to CA3 and Sub in DS, suggesting a delay in CA1 pyramidal development in the postnatal stages of DS. Although the functional consequences of a poorly developed CA1 PL are unknown, it suggests a less well-developed hippocampus and entorhinal cortex memory connectome in DS.

### 4.3. Postnatal Hippocampal Pyramidal Cell Maturation in DS

Postnatal neuronal maturation/differentiation is accompanied by an increase in type IV Nfls [44]. Nfls are cytoskeletal polymers responsible for the development and maintenance of neuronal morphology, playing an essential role in neurotransmission [45,46]. Using an antibody against non-phosphorylated Nfl heavy subunits (SMI-32), we found a similar pattern of non-phosphorylated Nfl-positive profiles in the postnatal hippocampus in DS and NTD cases. SMI-32 was observed in multipolar neurons in the hilus and pyramidal neurons in CA2 and Sub in both the youngest NTD and DS cases. Comparable results were reported for the hippocampus and Sub during the postnatal development of 3-week-old rhesus monkeys (*Macaca mulatta*) [47] suggesting early maturation of hippocampal complex neurons is conserved across primate species. NTD pyramidal Sub neurons were immunopositive for non-phosphorylated Nfl heavy subunits in the basal dendrites at the youngest ages compared to DS. The oldest cases in both groups revealed stronger SMI-32 immunoreactivity in CA1 field compared to the younger cases indicating a regional and temporal expression of non-phosphorylated Nfl heavy subunits in the postnatal hippocampus, most likely associated with increased neural transmission and improvement of hippocampal memory-related skills during infancy and adolescence.

SMI-34-ir fibers were seen in the alveus and, lacunosum moleculare stratum while only a few in the stratum radians and hilus. At 60 weeks, SMI-34 positive fibers were observed in NTD, but this was not observed until 83 weeks in DS. SMI-34 fibers were reported in young adult male Sprague Dawley rats within the mossy fibers of CA2, CA3, and the hilus, but absent in the GL and ML of the DG [48]. The delay in the presence of SMI-34 in the hippocampal fibers could lead to slower hippocampal synaptic transmission in DS.

### 4.4. Postnatal Hippocampal CBP in DS

Alterations in GABAergic systems occur during brain development and extend to adulthood and are implicated in memory impairment in trisomic mouse models of DS [49]. In fact, GABAergic neurons containing CBP (e.g., Calb, Parv and Calr) form distinct subpopulations in the human and rodent hippocampus [50], which control neural network excitability and long-term potentiation, a mechanism underlying learning and memory [51,52,53,54]. Here, we found that the distribution of Calb immunostaining in the postnatal hippocampus was comparable between the youngest and the oldest postnatal NTD and DS cases. Calb-ir was strongest in DG granule cells, non-pyramidal cells in the CA1 field, and CA3 mossy fibers, but less reactive in the CA2 field in both groups. A similar distribution was reported in the hippocampus of five human cases aged 47–100 years where moderate Calb-ir was seen in CA2, mossy fibers surrounding CA3 neurons, stronger in CA1 non-pyramidal cells, and only light reactivity in CA1 pyramidal cells [50]. Altogether, these findings showed that Calb-ir in the hippocampus is similar between postnatal and adulthood ages as well as between DS and NTD postnatal cases.

Parv found in GABAergic neurons modulates short-term synaptic plasticity [55]. While in the adult hippocampus non-pyramidal Parv positive cells were observed in the hilus, CA1 and CA2 fields and Sub, we did not observe positive cells in the CA subfields, DG or Sub in DS or in the NTD postnatal cases. Conversely, we did find Parv-ir cells in the entorhinal cortex in a 196-week-old DS case (data not shown). Although De Lecea and colleagues [56] found Parv mRNA-expressing cells in the CA2/3 regions by postnatal day 10 in Sprague Dawley rats, it is important to note the developmental process of rodents is increased compared to humans [57]. Functionally, hippocampal CA1 interneurons containing Parv play a role in social memory discrimination [58]. When provided with Face-n-Food images, individuals with DS (between the ages of 9–18) had a harder time reporting faces compared to NTD cases and individuals with autistic spectrum disorder and Williams-Beuren syndrome, a genetic growth-delaying disorder [59]. These observations suggest that deficits in facial memory recognition are associated with alterations in CA1 Parv interneurons later in childhood in DS and other disorders. The absence of hippocampal Parv positive cells in postnatal DS and NTD cases (present findings) suggests that this neuronal phenotype occurs later during development.

Calr is another marker of GABAergic interneurons and is involved in intracellular calcium buffering and apoptosis [60]. While we found Calr-ir cells in different regions in the postnatal DS hippocampus, there were virtually no positive Calr-ir cells in the NTD hippocampus. Specifically, Calr-ir cells were observed in the GL of the DG, hilus and interneurons in the CA2/3 and CA1 fields at all ages in DS. Similar findings were observed in fetuses of 17–21 weeks of gestation, where the percentage of Calr positive neurons in the hippocampal GL and CA1 field, as well as in the fusiform gyrus and inferior temporal gyrus, was greater in DS than their NTD counterparts [61]. Moreover, we found Calr-ir fibers in the DG ML, lacunosum-moleculare, radians and oriens strata which were more abundant in postnatal DS than in NTD. Taken together, the presence of more Calr-ir profiles in the prenatal and postnatal hippocampus suggests an upregulation of GABAergic circuit activity that compromise the balance between excitatory-inhibitory synapses in the hippocampus in DS. These data support the hypothesis that GABAergic over-inhibition may cause synaptic and circuit disorganization leading to intellectual disabilities early in life in DS and extend to adulthood [62]. While a large body of evidence has shown that GABAergic drugs are pharmacologically effective in improving behavior in animal models of DS, clinical trials using GABAergic drugs have failed to meet cognitive endpoints in patients with DS [63], which led to the discontinuation of these trials. Based on these data, more research is required to better understand the role of the GABAergic system dysfunction during brain development as a therapeutic approach for the treatment of cognitive impairment in DS.

### 4.5. Postnatal Hippocampal APP/Aβ_1–42_ and Tau in DS

Although many triplicated genes may be involved in the developmental defects due to trisomy 21, *APP* overexpression appears to be the determinant of many neurodevelopmental alterations that characterize DS, including the fate of neural precursor cells [20,24]. The overexpression of *APP* [64] and its catalytic toxic product Aβ_1–42_ occurs during prenatal DS brain development [25,65]. In this regard, a previous study reported higher levels of APP in the hippocampal stratum oriens in prenatal DS, possibly causing disruption morphogenesis [25]. Although little is known about APP deposition during postnatal hippocampal development, we found sporadic diffuse deposits of APP/Aβ in all DS and NTD cases. In contrast, Aβ_1–42_ immunostaining was not detected in the hippocampus in the same cases. Interestingly, we have also found APP/Aβ positive deposit in the postnatal cerebellar and frontal cortex in DS and NTD cases as well as in the hippocampus and frontal cortex of 12- and 24-year-old NTD cases, respectively (see Appendix A). Conversely, Davidson and colleagues [7] described amyloid diffuse deposits in the temporal cortex of a 13-year-old but not in neonates. The discrepancy between these findings and the current finding of APP/Aβ diffuse plaques in DS and NTD may be related to differences in the antibodies. Davidson et al. [7] used 4G8 (epitope Aβ_18–22_ residues), which detects human Aβ species, while 6E10 (epitope Aβ_1–16_ residues) recognizes human Aβ and APP species (APPα). Validation of the staining observed with the 6E10 antibody was confirmed by the lack of immunoreaction after antibody preabsorption with its peptide as well as deletion of the primary and/or secondary antibodies using tissue from postnatal NTD and DS. Together, these immunohistochemical control experiments lend support to specificity of APP/Aβ positive bursts seen in the present study in the postnatal hippocampus in NTD and DS. The data also suggest that these parenchymal accumulations are comprised of non-amylodogenic APP species. Further conformation and characterization of the APP/Aβ staining reported in the prenatal hippocampus is required. Functionally, APP has been implicated in cell-cell interaction, cell-substrate adhesion, neuronal differentiation, and synaptogenesis [66]. However, mutations of the *APP* gene are known to contribute to the early onset of dementia in familial AD (FAD) and is key to Aβ plaque formation and neurodegeneration in sporadic and FAD [67,68]. Since there was no apparent detection of Aβ_1–42_ immunoreactivity in the postnatal hippocampus in either group, and APP/Aβ loads were similar between groups, we suggest that APP is associated with the postnatal differentiation of the hippocampus in both groups. However, whether the overexpression of prenatal *APP* alone or in conjunction with other genes located on chromosome 21, such as *DYRK1A, RCAN1* and *OLIG1/2,* play a role in postnatal hippocampus development needs further investigation.

P-tau is a main component of NFTs in AD [69,70]. Individuals with DS beyond 40 years of age display both amyloid plaques and tau-bearing NFTs [8,71]. The shortest transcript of tau termed fetal tau or N0R3 is found in both the normal and DS fetal brain during development [72]. N0R3 is phosphorylated around Ser202 and detected in long axonal tracts, peaking in the human brain at mid-gestation [26,73]. In fact, prenatal tau was associated with axonal guidance, particularly with axonal growth and targeting [74], which is downregulated prior to birth in humans [26]. In accordance with these findings, we found p-tau (CP-13) in fiber tracts in the hippocampus at 20 and 22 gestational weeks in DS and NTD, respectively (Figure 5A,B); however, p-tau (CP-13, PHF-1 or AT8) was lacking in all areas of the postnatal hippocampus at all ages in DS and NTD. A reduction in p-tau at 16 to 28 weeks of gestation was associated with tau dysfunction, but not axonal depletion [27]. A recent imaging study using the tau tracers ^3^H-THK5117 and ^3^H-MK6240 demonstrated binding in developing DS fetal cortex, but not in control cases [75]. Taken together, these observations suggest that occurrence of fetal tau disturbances in DS lead to connectome modifications in development and adulthood.

## 5. Conclusions

In sum, we found poor cellular organization and morphology of the neurons in the postnatal DS hippocampus, specifically in the CA1 subfield. The distribution of SMI-32-ir neurons in the hilus, CA subfields and Sub were comparable in all DS and NTD cases. However, pyramidal apical and basal dendrites displayed more intense SMI-32 immunoreactivity in NTD. The distribution of Calb-ir neurons was similar between the youngest and the oldest NTD and DS cases. Parv immunoreactivity was not detected in the postnatal hippocampus at any age in both groups. By contrast, numerous Calr-ir cells and fibers were observed in the youngest DS samples, while Calr-ir fibers only were seen in NTD cases. There were no differences in hippocampal APP/Aβ plaque loads between DS and NTD, but p-tau was not detected in any case. DG DCX-ir cells declined with age in both DS and NTD, but lesser DCX-ir cells were observed after 61 weeks in DS. We also found a stronger negative correlation between DG DCX-ir cell number and brain weight in DS, while age and brain weight showed a stronger positive correlation in NTD. These findings suggest that deficits in hippocampal neurogenesis/migration and GABAergic neuronal inhibitory systems occur with lower brain weight in early postnatal developmental that contribute to cognitive impairments in DS. Finally, it is important to acknowledge caveats associated with this study. The small number of cases examined is always a problem in developmental studies using human postmortem tissue from postnatal NTD and DS infants and toddlers since these cases are uncommon. Therefore, the present findings need to be confirmed with a larger number of cases and interpreted with caution. For those interested in learning more about brain donation for DS research click this link: https://www.brightfocus.org/alzheimers/grant/international-brain-bank-down-syndrome-research. 

## Figures and Tables

**Figure 1 jcm-10-03414-f001:**
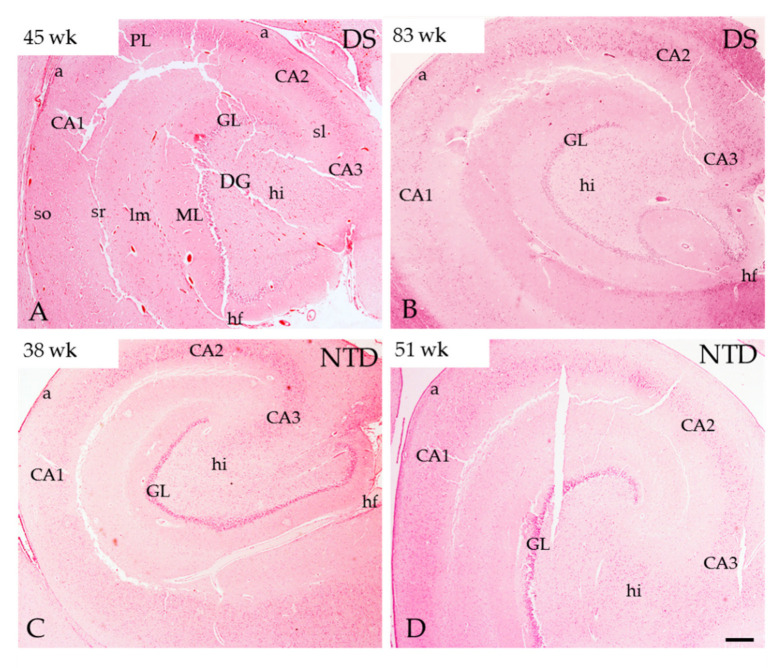
H&E-stained sections of postnatal caudal hippocampus in DS and NTD cases. Abbreviations: a = alveus, CA1 = hippocampal subfield CA1, CA2 = hippocampal subfield CA2, CA3 = hippocampal subfield CA3, GL = granule cell layer, hf = hippocampal fissure, hi= hilus, lm= stratum lacunosum-moleculare, ML = molecular layer, PL = pyramidal layer, sl = stratum lacunosum, so = stratum oriens, sr = stratum radiatum. Scale bar in (**D**) = 500µm and applies to panels (**A–C**).

**Figure 2 jcm-10-03414-f002:**
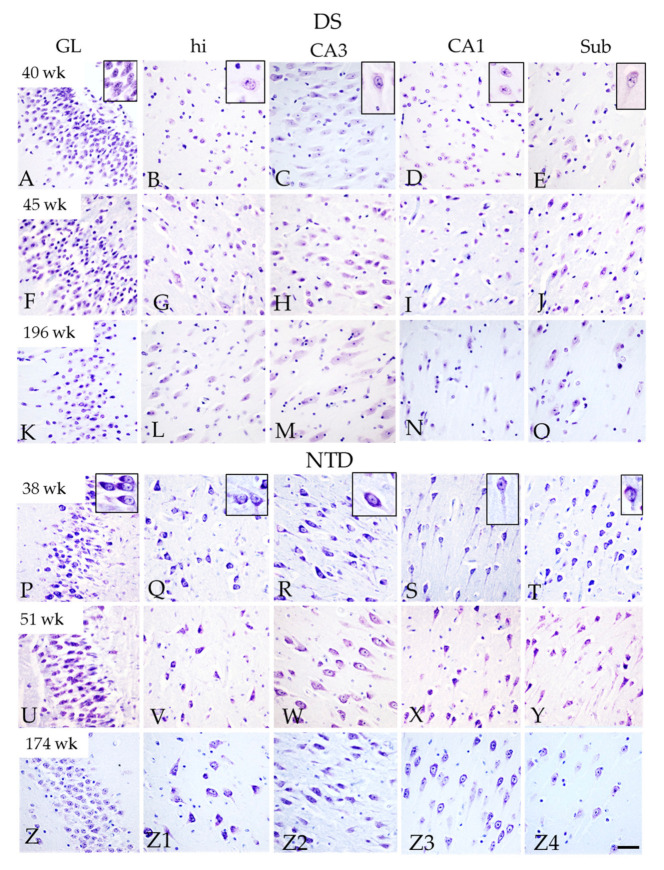
Thionin-stained sections showing differences in the morphological characteristics of neurons in the GL, dentate hilus, CA3, CA1, and Sub in a 40 (**A**–**E**), 45 (**F**–**J**), 196 week-old (**K**–**O**) DS and a 38 (**P**–**T**), 51 (**U**–**Y**), and 174-week-old (**Z**–**Z4**) NTD case. High magnification of thionin-stained neurons are shown in (**A**–**E**) and (**P**–**T**) insets. Note the better cellular organization and stronger soma and dendritic staining in all the areas in NTD compared to DS. Abbreviations: CA1 = hippocampal subfield CA1, CA3 = hippocampal subfield CA3, GL = granule cell layer, hi = hilus, Sub = subiculum. Scale bar in (**Z4**) = 50 µm and applies to (**A**–**Z3**) and 15 µm for all insets.

**Figure 3 jcm-10-03414-f003:**
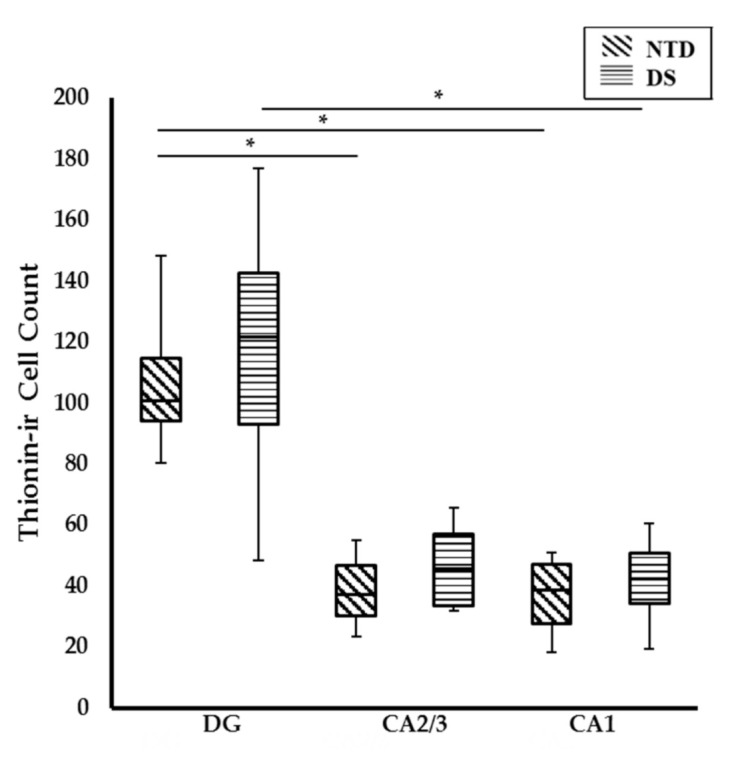
Box-plot showing significantly higher numbers of thionin stained cells in the DG compared to CA1 (*p* = 0.004) and CA2/3 (*p* = 0.043) subfields in NTD and DG compared to CA1 in DS (*p* = 0.003). No significant difference was found between groups. * *p* < 0.05.

**Figure 4 jcm-10-03414-f004:**
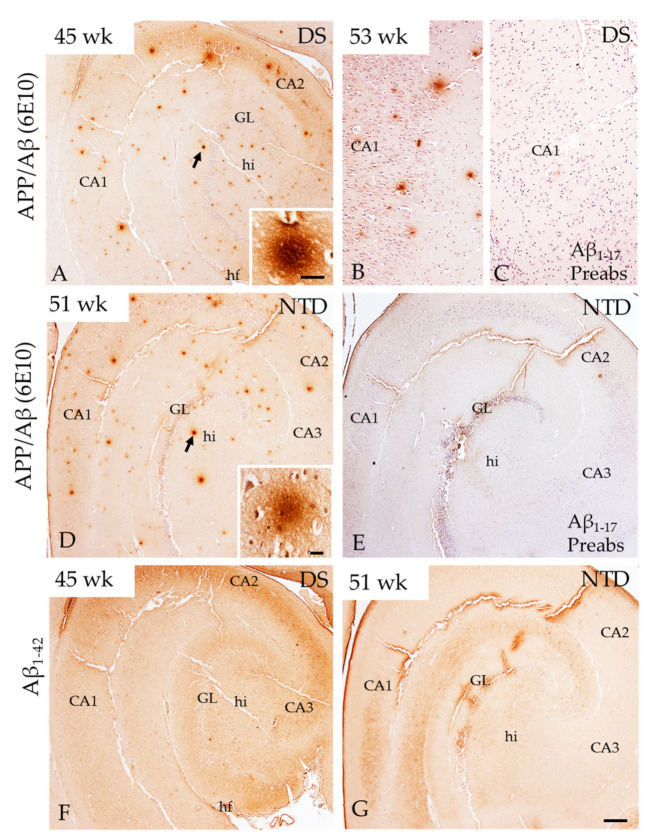
Images of APP/Aβ (6E10) (**A,B,D**) and Aβ_1–42_ (**F,G**) immunopositive profiles in the hippocampus at postnatal week 45 and 53 in DS, and at week 51 in a NTD subject. High magnification images (insets) of APP/Aβ-ir profiles (arrows) in DS (**A**) and NTD (**D**). Note the random distribution of APP/Aβ-ir and absence of Aβ_1–42_-ir positive deposits in the hippocampus in both DS (**A**) and NTD (**D**). Additionally, absence of APP/Aβ-ir was noted in the hippocampus at postnatal week 53 (**C**) compared to antibody staining in panel B and similarly at 51 week in panel (**E**) compared to (**D**) in NTD (**E**) after antibody preabsorption with the Aβ_1–17_ peptide. Sections in (**B**,**C**,**E**) were counterstained with hematoxylin. Abbreviations: CA1 = hippocampal subfield CA1, CA2 = hippocampal subfield CA2, CA3 = hippocampal subfield CA3, fi = hippocampal fimbria, GL = granule cell layer, hi = hilus, hf = hippocampal fissure, Aβ_1–17_ Preabs= preabsorption with Aβ_1–17_ peptide, Sub = subiculum. Scale bar in (**G**) = 500 µm applies to (**A**,**D**–**F**) and in (**B**) and (**C**) = 100 µm; insets: (**A**) = 20 µm and (**B**) = 25 µm.

**Figure 5 jcm-10-03414-f005:**
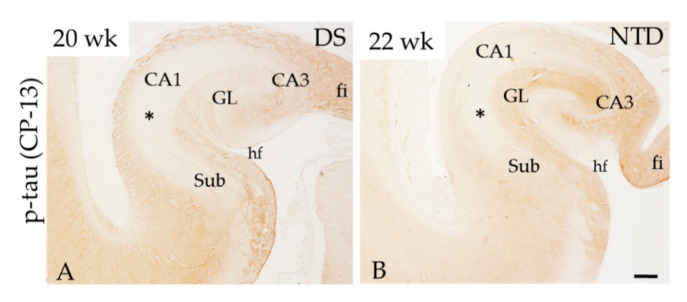
Low-power images showing p-tau (CP-13) labeling at DS postnatal week 20 (**A**) and 22 in a NTD (**B**) case. Note that fiber tracts but not cellular regions (*) of the hippocampus were CP-13 (tau) immunopositive. Abbreviations: CA1 = hippocampal subfield CA1, CA2 = hippocampal subfield CA2, CA3 = hippocampal subfield CA3, fi = hippocampal fimbria, GL = granule cell layer, hi = hilus, hf = hippocampal fissure, Sub = subiculum. Scale bar in (**B**) = 600 µm applies (**A**).

**Figure 6 jcm-10-03414-f006:**
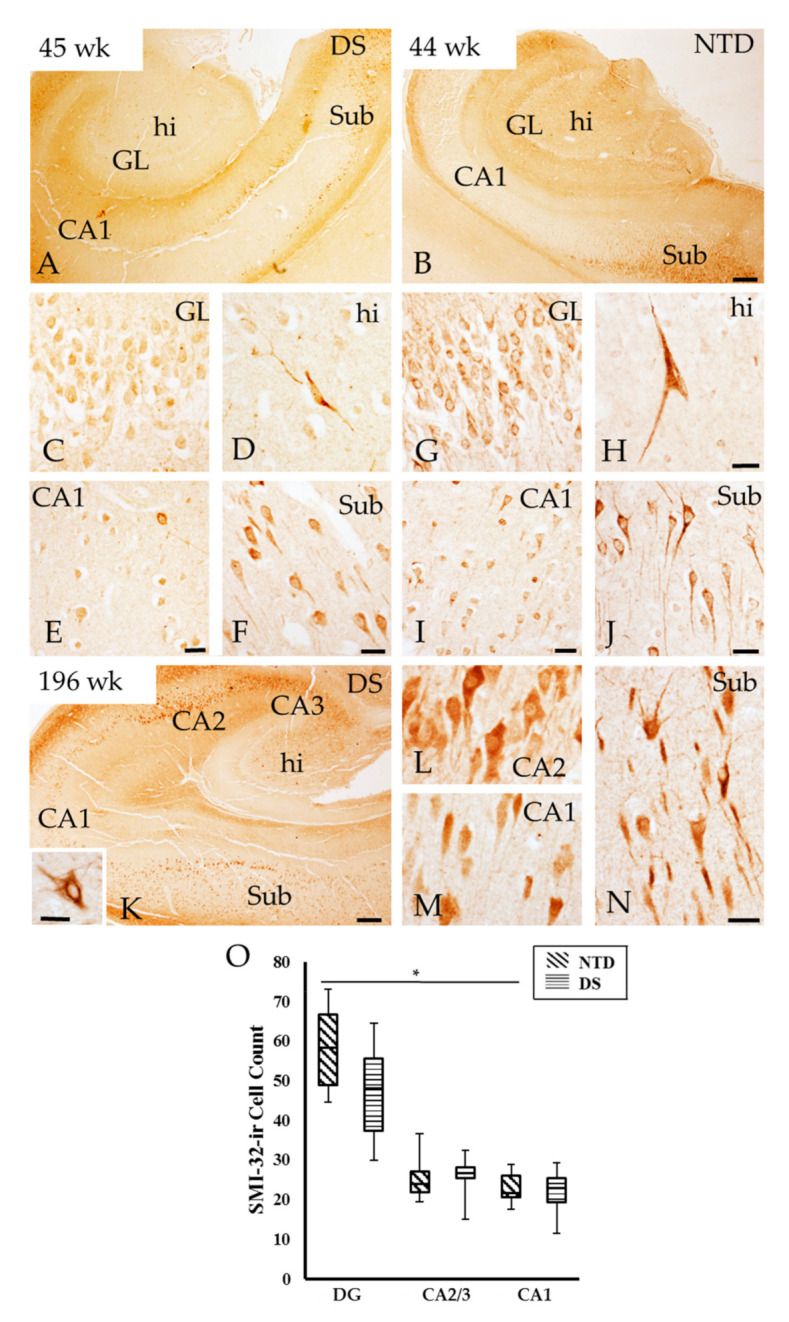
Distribution of SMI-32 immunoreactivity in the hippocampus of a 45 (**A**) and 196 (**K**) postnatal week DS and 44 week NTD case (**B**). Strong SMI-32-ir cells were observed in the hilus (**D**,**H**) and Sub (**F,J**) compared to the GL (**C,G**) and CA1 subfield (**E,I**) in both cases. CA2 (**L**), CA1 (**M**), and Sub (**N**) pyramidal and hilar cells ((**K**) inset) were SMI-32 positive at postnatal week 196 in DS. Note stronger cellular SMI-32 immunostaining in the oldest compared to the youngest DS and NTD cases. Abbreviations: CA1 = hippocampal subfield CA1, CA2 = hippocampal subfield CA2, CA3 = hippocampal subfield CA3, DG = dentate gyrus, GL = granule cell layer, hi = hilus. Scale bars in (**B**) and (**K**) = 500 µm and applies to (**A**), in (**H**) = 25 µm and applies to (**C**,**D**,**G**), in (**E**,**F**,**I**,**J**) = 25 µm, in (**N**) = 25 µm and applies to (**M**) and (**L**). Scale bar in the panel (**K**) inset = 25 µm. (**O**), Box-plot showing significantly higher numbers of SMI-32-ir cells in the DG compared to CA1 subfield in NTD (*p* = 0.004). No significance differences were found between groups. * *p* < 0.05.

**Figure 7 jcm-10-03414-f007:**
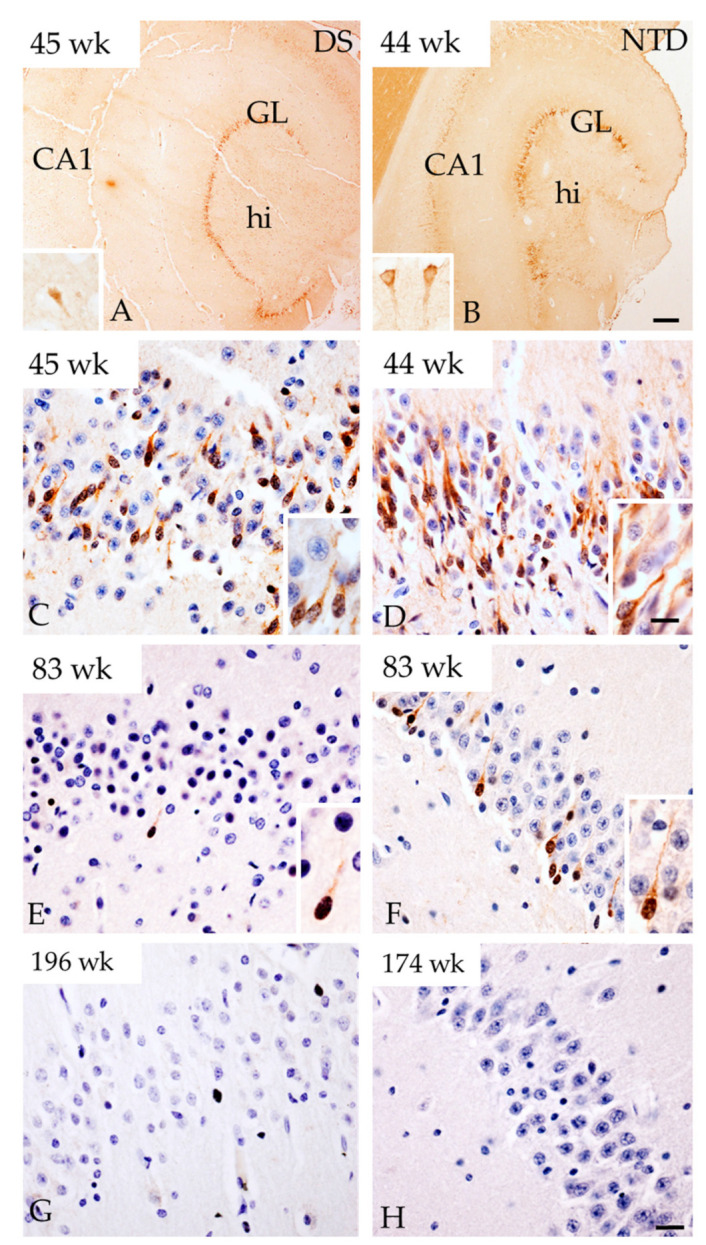
Images showing strong DCX immunoreactivity in the DG GL at week 45 in DS (**A**) and at 44 weeks in NTD (**B**). Insets in panel (**A**) and (**B**) show DCX-ir CA1 pyramidal cells in a DS 45 and NTD 44 week cases, respectively. Images showing DCX-ir cells in the GL at postnatal weeks 45, 83 and 196 in DS and at 44, 83, 174 weeks in NTD cases. Insets in panels (**C**–**F**) show details of DCX positive cells in the GL. Note that less cells were positive for DCX in DS at 45 (**C**) and 83 (**E**) postnatal weeks compared to 44 (**D**) and 83 (**F**) weeks in NTD and were absent at 196 (**G**) and 174 (**H**) weeks in DS and NTD, respectively. Section in (**C**–**H**) were counterstained with hematoxylin. Abbreviations: CA1 = hippocampal subfield CA1, GL = granule cell layer, hi = hilus. Scale bar in (**B**) = 500 µm and applies to (**A**), and in (**H**) = 25 µm applies to (**C**–**G**) panels, in (**D**) inset = 10 µm and applies to insets of (**A**–**F**).

**Figure 8 jcm-10-03414-f008:**
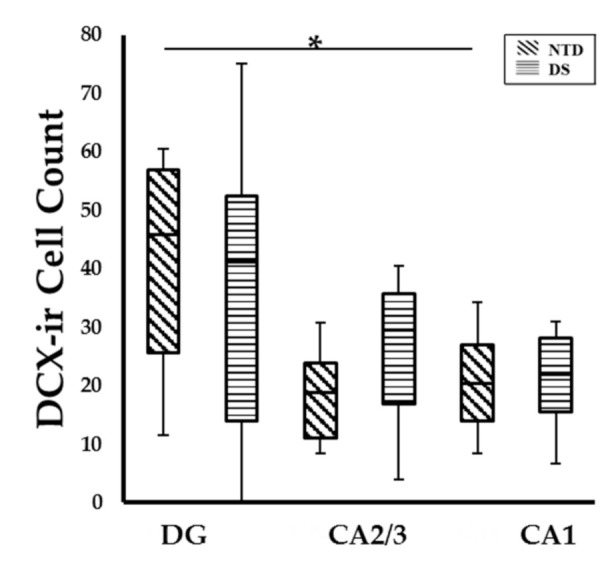
Box-plot showing no significant difference in DCX cell count between groups, however significantly higher numbers of DCX-ir cells were observed in the DG compared to CA1 subfield (*p* = 0.031) in NTD. * *p* < 0.05.

**Figure 9 jcm-10-03414-f009:**
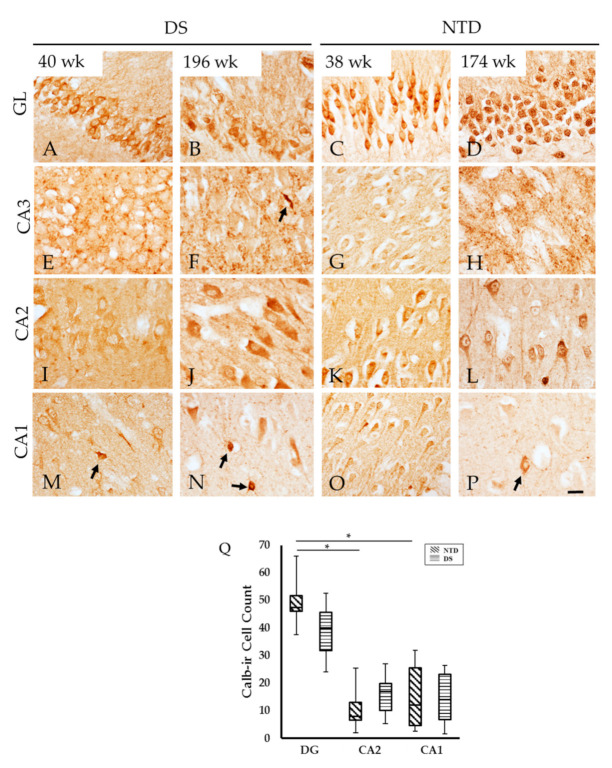
Images showing Calb-ir cells in the GL, CA2 and CA1 subfields, as well as Calb-ir fibers in the CA3 subfield at 40 (**A**,**E**,**I**,**M**) and 196 (**B**,**F**,**J**,**N**) postnatal weeks in DS and 38 (**C**,**G**,**K**,**O**) and 174 (**D**,**H**,**L**,**P**) week in NTD cases. Note that at the older ages (**N**,**P**), pyramidal cells in the CA1 display pale immunostaining compared to the younger (**M**,**O**) cases in both groups. Granule cells in the DG were highly Calb-ir in both groups (**A**–**D**) and few small Calb-ir interneurons were observed in CA3 (**F**) and CA1 (**M**,**N**,**P**) subfields (arrows). Q, Box plot showing no significant difference in Calb cell count between groups, however, there were significantly higher numbers of Calb-ir cell observed in the DG compared to CA2 (*p* = 0.001) and CA1 subfields (*p* = 0.033) within the NTD group. Abbreviations: CA1 = hippocampal subfield CA1, CA2 = hippocampal subfield CA2, CA3 = hippocampal subfield CA3, GL = granule cell layer. Scale bar in (**P**) = 25µm and is applied to (**A**–**O**).

**Figure 10 jcm-10-03414-f010:**
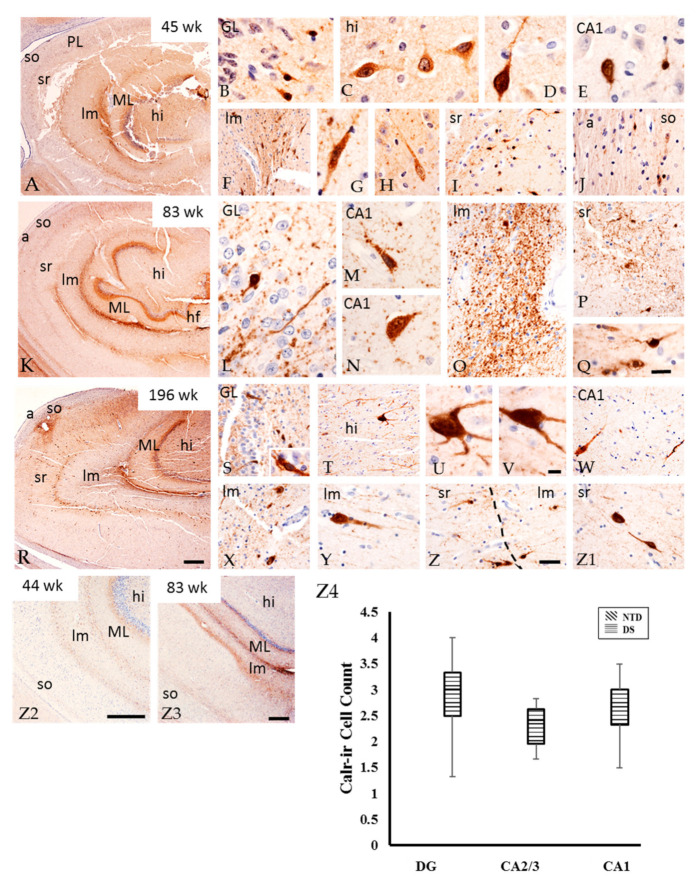
Images showing the distribution Calr-ir profiles in the DG ML, lacunosum-moleculare, radiatum and oriens strata in a 45 (**A**), 83 (**K**), 196 (**R**) postnatal week-old DS cases, as well as in a 44 (**Z2**) and 83 (**Z3**) week-old NTD case. Note an increase in the number of Calr-ir profiles in the oldest DS case (**R**) and less Calr immunostaining in these layers in NTD (**Z2**,**Z3**) cases. Higher magnification images showing Calr immunostaining in small cells in the GL (**B**,**L**,**S**), multipolar cells in the hilus (**C**,**D**,**T**–**V**), non-pyramidal CA1 subfield (**E**,**M**,**W**), and elongated cells in the stratum lacunosum-moleculare (**F**–**H**,**O**,**X**–**Z**). Calr-ir fibers were seen coursing within the stratum lacunosum-moleculare (**F**,**O**,**X**), radiatum (**I**,**P**,**Z**) and oriens/alveus (**J**) in a 45 (**B**–**J**), 83 (**L**–**Q**), and 196 (**S**–**Z1**) postnatal week-old DS case. All sections were counterstained with hematoxylin. Scale bar in (**R**) = 500 µm and applies to (**A**) and (**K**), in (**Z2**) and (**Z3**) = 500 µm, in (**Z**) = 25 µm and applies to (**F**,**I**,**J**,**O**,**P**,**S**,**T**,**W**,**X**,**Y**) and (**Z1**), in (**Q**) = 10 µm and applies to (**B**,**C**,**D**,**E**,**G**,**H**,**L**,**M**,**N**), in (**V**) = 10 µm and applies to (**U**) and inset (**S**). (**Z4**), Box-plot showing no significant difference in Calr-ir cell count within DS cases. Abbreviations: a = alveus, CA1 = hippocampal subfield CA1, CA2 = hippocampal subfield CA2, CA3 = hippocampal subfield CA3, GL = granule cell layer, hf = hippocampal fissure, hi = hilus, lm = stratum lacunosum-moleculare, ML = molecular layer, PL = pyramidal layer, so = stratum oriens, sr = stratum radiatum.

**Figure 11 jcm-10-03414-f011:**
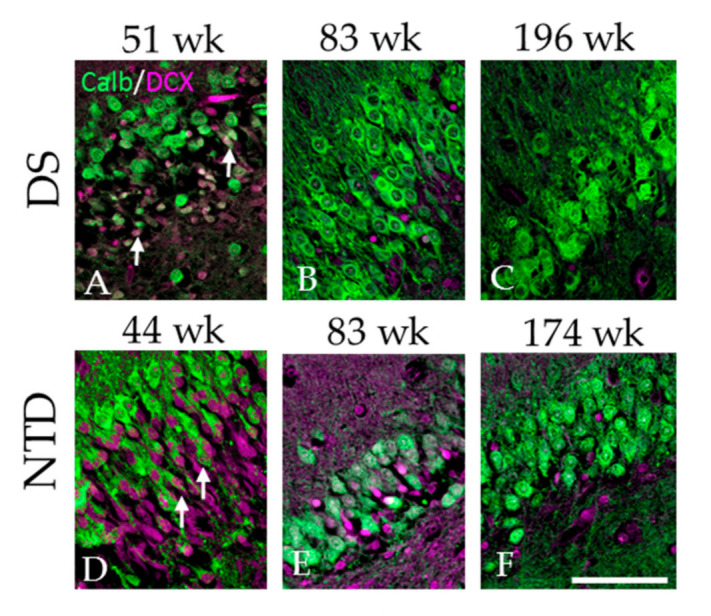
Immunofluorescent images of the DG granule cell layer showing single DCX labeled cells (pink) and Calb (green) and double DCX/ Calb positive cells (pink/green) in a 51 (**A**), 83 (**B**) and 196 (**C**) week-old DS case and a 44 (**D**), 83 (**E**) and 174 (**F**) postnatal week-old NTD case. Note that double labeled DCX + Calb positive cells (arrows) were mainly observed in the youngest 51 DS (**A**) and 44 week-old (**D**) NTD cases. Scale bar = 70 µm.

**Figure 12 jcm-10-03414-f012:**
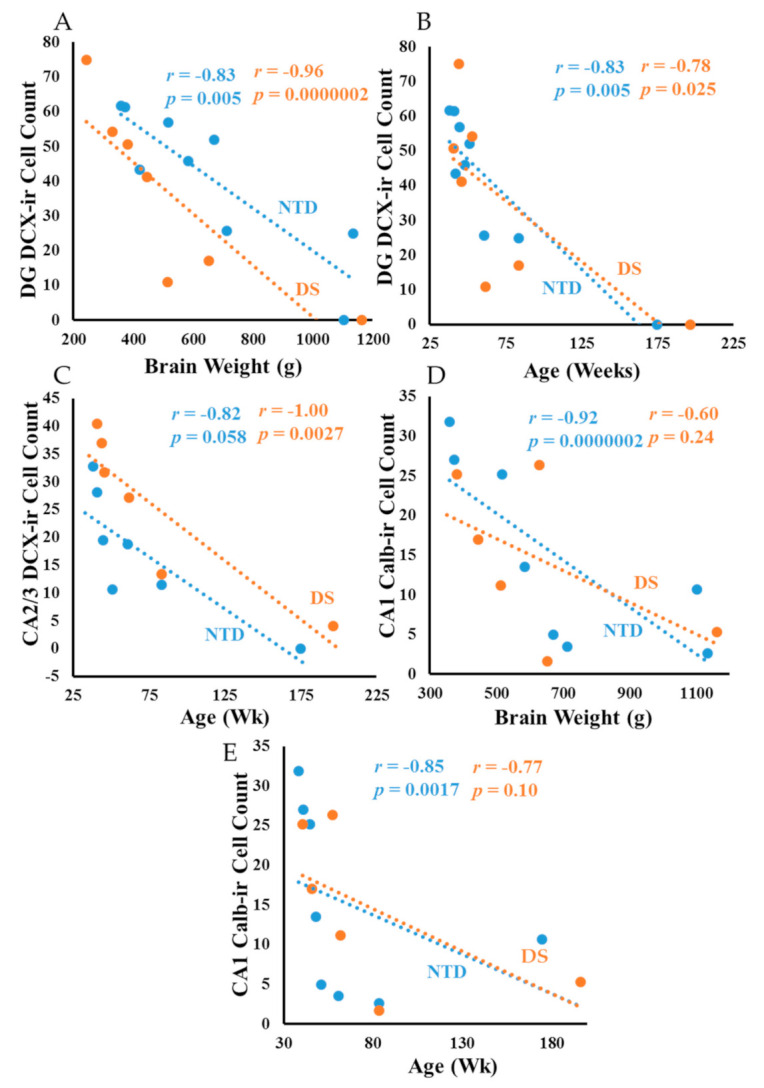
Linear regression analysis showing a strong negative correlation between DG DCX-ir cell counts and brain weight in DS (**A**); Spearman rank, *r* = −0.96 *p* = 0.0000002), but weaker in NTD (**A**); Spearman rank, *r* = −0.83 *p* = 0.005). DG DCX-ir cell counts correlated negatively with age in NTD (**B**); Spearman rank, *r* = −0.83 *p* = 0.005) and DS (**B**); Spearman rank, *r* = −0.78 *p* = 0.025). DS CA2/3 DCX cell counts correlated negatively with age (**C**); Spearman rank, *r* = −1.00 *p* = 0.0027), but not in NTD. By contrast, CA1 Calb-ir cell counts showed a negative correlation with brain weight (**D**); Spearman rank, *r* = −0.92 *p* = 0.0000002) and age (**E**); Spearman rank, *r* = −0.85 *p* = 0.0017) in NTD.

**Figure 13 jcm-10-03414-f013:**
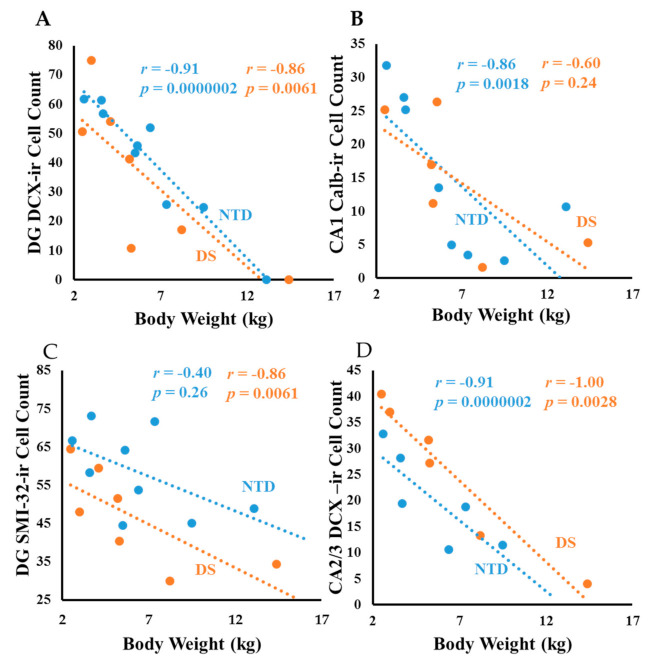
Linear regression analysis showing negative correlations between body weight and DCX DG (**A**); Spearman rank; *r* = −0.91 *p* = 0.0000002), and Calb-ir CA1 (**B**); Spearman rank; *r* = −0.86 *p* = 0.0018) counts in NTD, while correlations between body weight and DG DCX (**A**); Spearman rank; *r* = −0.86 *p* = 0.0061), DG SMI-32 (**C**); Spearman rank; *r* = −0.86 *p* = 0.0061), CA2/3 DCX (**D**); Spearman rank; *r* = −1.0 *p* = 0.0028) neuron numbers were less statistically significant in DS.

**Figure 14 jcm-10-03414-f014:**
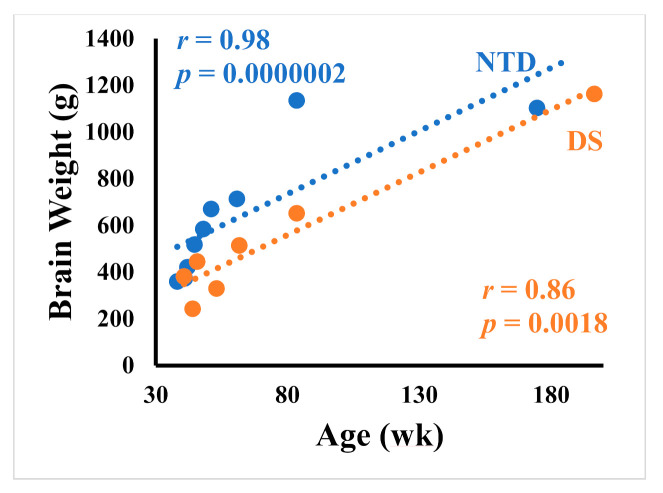
Linear regression analysis showing a stronger positive correlation between age and brain weight in the NTD compared to DS (Spearman rank, NTD *r* = 0.98 *p* = 0.0000002, DS *r* = 0.85 *p* = 0.0017).

**Table 1 jcm-10-03414-t001:** Case characteristics.

	C.D.	Sex	Age at Birth (wk)	Age at Death (wk)	Post-natal Life (wk)	Brain Weight (g)	Body Weight (kg)	Height (cm)	PMI (h)	Tissue Source	Cause of Death/Comorbidity
1	DS	M	31.7	40.7	9	381.0	2.5	49	58	LCH	congenital heart disease
2	DS	F	40.0	44.0	4	243.0	3	49.3	20	LCH	congenital heart disease
3	DS	F	37.0	45.7	8.5	445.0	5.2	55	16	LCH	congenital heart disease
4	DS	M	38.0	53.0	15	330.0	4.1	49	21	LCH	lung disease
5	DS	M	40.0	57.4	17.2	630.0	5.6	48	14	PCH	congenital heart disease
6	DS	F	40.0	61.7	21.7	514.0	5.3	60	34	LCH	acute pneumonia
7	DS	M	40.0	83.5	23.5	652.0	8.2	84.3	10	PCH	congenital heart disease
8	DS	M	40.0	196.4	156.4	1163.0	14.4	97	22	LCH	lung damage
9	NTD	F	38.1	38.1	0	360.0	2.6	48.5	--	PCH	pulmonary hemorrhage
10	NTD	F	37.0	41.0	4	373.3	3.6	55.5	6	PCH	congenital heart disease
11	NTD	F	39.0	42.0	3	420.9	5.52	39	16	PCH	congenital heart disease
12	NTD	M	40.0	44.7	4.7	518.0	3.7	54.5	26	PCH	congenital heart disease
13	NTD	M	39.0	48.0	54	584.2	5.7	55.8	26	PCH	congenital heart disease
14	NTD	M	40.0	51.0	11	670.0	6.4	63.5	17	PCH	septicemia
15	NTD	F	27.0	60.8	33.7	713.5	7.4	70.2	--	PCH	systemic inflammatory response syndrome
16	NTD	M	40.0	83.5	23.54	1134.9	9.5	77.8	29	PCH	acute pneumonia
17	NTD	M	40.0	174.7	135	1103.0	13.1	90.1	18	PCH	lymphoma

PMI: postmortem interval, C.D.: Clinical Diagnosis; LCH: Ann & Robert H. Lurie children’s hospital of Chicago; PCH: Phoenix children’s hospital.

**Table 2 jcm-10-03414-t002:** Antibody description and dilution.

	Description	Dilution	Company
SMI-32	Mouse monoclonal purified anti-neurofilament H, non-phosphorylated antibody	1:500	Biolegend, San Diego, CA, USA
SMI-34	Mouse monoclonal purified anti-neurofilament H, phosphorylated antibody	1:200	Biolegend, San Diego, CA, USA
DCX	Neuronal migration protein Doublecortin (E-6)	1:250	Santa Cruz Biotechnology, Dallas, TX, USA
Ki-67	Monoclonal mouse anti-human Ki-67 antigen clone MIB-1	1:200	Dako, Denmark A/S, Glostrup, Denmark
Calb	Rabbit polyclonal anti-calbindin D-28K	1:500	Swant, Marly, Switzerland
Parv	Mouse monoclonal anti-parvalbumin	1:500	Millipore, Billerica, CA, USA
Calr	Rabbit anti-calretinin	1:500	Millipore, Billerica, CA, USA
APP/Aβ (6E10)	Mouse monoclonal against human β-Amyloid (Aβ, 1–16 aa) and APP	1:300	BioLegend, San Diego, CA, USA
Aβ_1–42_	Rabbit polyclonal anti-Aβ_1–42_	1:100	Millipore, Billerica, CA, USA
CP-13	Phospho-tau (Ser202) mouse monoclonal antibody	1:100	gift from Peter Davies
PHF-1	Phospho-tau (Ser396/Ser404) mouse monoclonal antibody	1:100	gift from Peter Davies
AT8	Phospho-tau (Ser202, Thr205) mouse monoclonal antibody	1:100	Invitrogen, Carlsbad, CA, USA

**Table 3 jcm-10-03414-t003:** Summary of case demographics.

	NTD (*n* = 9)	DS (*n* = 8)	*p*-Value
Age (wk)	64.87 ± 14.49 * Min. 50.38; Max. 79.36	72.80 ± 18.29 * Min. 54.51; Max. 91.09	*p* > 0.05 ^a^
Brain Weight (g)	653.08 ± 97.15 * Min. 555.93; Max. 750.23	544.75 ± 101.40 * Min. 443.35; Max. 646.15	*p* > 0.05 ^a^
Body Weight (kg)	6.38 ± 3.29 * Min. 2.59; Max. 13.10	6.036 ± 3.81 * Min. 2.50; Max. 14.40	*p* > 0.05 ^a^
Height (cm)	61.66 ± 15.64 * Min. 39; Max. 90.10	61.45 ± 18.78 * Min. 48; Max. 97	*p* > 0.05 ^a^
PMI (h)	19.71 ± 2.99 * Min. 16.72; Max. 22.70	24.37 ± 5.41 * Min. 18.96; Max. 29.780	*p* > 0.05 ^a^
Gender (%)	Male (55.55%); Female (44.44%)	Male (62.50%); Female (37.50%)	*p* > 0.05 ^b^

* Mean ± Standard Error (SE); ^a^: Non-parametric Mann—Whitney rank sum test; ^b^: Fisher exact test.

## Data Availability

Original data are available from the corresponding author upon reasonable request.

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
