# Peer review of "Postnatal Cytoarchitecture and Neurochemical Hippocampal Dysfunction in Down Syndrome"

_jcm, 2021, doi:10.3390/jcm10153414_

Round 1

Reviewer 1 Report

The paper is strong despite the small 'n'. There are just a few editorial issues to consider. First, the use of abbreviations needs to be tightened up. Some abbreviations are used in the text but only defined in the legends of the figures. Be sure that all abbreviations are defined the first time that they are used in the text. Second, the paper is very long, particularly, the results section. Please consider moving qualitative data with figures of stained sections to an addendum. Then, the results section can focus on quantitative data (i.e. box plots) while mentioning interesting observations made qualitatively. This would shorten the paper and focus the narrative on the most significant aspects of the research. Finally, there appears to be a typo on line 428. Figure 13A is a correlation with brain weight not body weight. This sentence should be corrected. Overall, the paper is publishable with few edits.

Author Response

Response to Reviewer 1 Comments

Point 1: First, the use of abbreviations needs to be tightened up. Some abbreviations are used in the text but only defined in the legends of the figures. Be sure that all abbreviations are defined the first time that they are used in the text.

Response 1: We thank the reviewer for this comment. In the revised manuscript, abbreviations are defined the first time they are used in the text.

Point 2: Second, the paper is very long, particularly, the results section. Please consider moving qualitative data with figures of stained sections to an addendum. Then, the results section can focus on quantitative data (i.e. box plots) while mentioning interesting observations made qualitatively. This would shorten the paper and focus the narrative on the most significant aspects of the research.

Response 2: We thank the reviewer for this suggestion. We moved figures 6, 8 (Tracings), 12 and 15 to the supplemental data section.

Point 3: Finally, there appears to be a typo on line 428. Figure 13A is a correlation with brain weight not body weight. This sentence should be corrected.

Response 3: Typo was corrected in the revised manuscript (see page 21, line 414).

Reviewer 2 Report

In this manuscript, the authors assess the neuronal composition of hippocampi from Down syndrome (DS) and age-matched neurotypically developing (NTD) controls ranging in age from 38 weeks gestation to 3 years.  The authors report differences in SMI-312 staining, altered ratios of doublecortin-positive neurons comparing dentate gyrus to CA1, and altered distribution of neuronal Calretinin in the DS brain.  The authors found no differences in Amyloid-beta-42 deposition or phospho-tau deposition in the DS brain compared to controls.  The authors conclude that the differences that they observe in the neuronal composition of the DS hippocampus contribute to cognitive impairment in DS. Descriptions of the postnatal developing brain are crucial for understanding functional impairment in DS, the authors have performed a thorough series of histologic and immunohistologic studies to address a number of important, interesting questions; however, a number of questions need to be addressed:

1) Did the authors control for where along the anteroposterior axis the hippocampi were sampled (e.g., at the level of the lateral geniculate nucleus or more anterior pes cortex)?  The hippocampus can look more or less disorganized based on where along the A/P axis it is sampled.

2) Were DS and NTD hippocampal sections stained simultaneously with Thionin to control for variation in staining intensity between sets of slides?  Were slides all stained simultaneously, blind to experimental group? 

3) With no significant difference between numbers of neurons in dentate gyrus or CA1-3, what is the significance of an altered ratio of dentate to pyramidal cell numbers comparing NTD and DS brain?  A similar observation was made in the SMI-32 and Doublecortin analyses, but the significance of these differences are not clear either.

4) In the DCX tracings (Figure 8), the DS hippocampus looks dramatically different than the age-matched NTD hippocampi depicted, yet the numbers of neurons are not significantly different between groups.  Do the authors think that there is a significant difference in the morphology of the hippocampus?  Was this apparent in every DS brain compared to every NTD brain? Is there a way in which a morphologic abnormality can be quantified?

5) The authors do not find Parv+ neurons in either DS or NTD cohorts. If they are absent in the NTD brains, why is their absence in the DS brain considered abnormal?  With no evidence of a difference in Parv-expressing cells between DS and NTD brain, the logic for considering a role for these neurons in cognitive impairment in DS is not clear.

6) Were calreticuin stains performed simultaneously under identical conditions between groups?  The NTD stains shown seem to have a lower background staining, suggesting the staining conditions are not identical.

7) What is the significance of a correlation between brain weight and DCX staining when it is also present in NTD brains?  It seems misleading to ignore that this correlation is present in both groups.

8) Linear regression analysis comparing DG DCX cell numbers does not show more rapid loss of DCX neurons in DS brain compared to NTD brain (Figure 13B).  Is there additional data supporting the difference stated in the discussion (page 29, lines 487-489)?  Why do the authors draw this conclusion?

9) Are diffuse amyloid plaques really present in the neonatal NTD brain? Could this be a staining artifact?  Have diffuse amyloid plaques been described in the NTD brain before?  This is a surprising result if it is real.

Author Response

Response to Reviewer 2 Comments

Point 1: Did the authors control for where along the anteroposterior axis the hippocampi were sampled (e.g., at the level of the lateral geniculate nucleus or more anterior pes cortex)?  The hippocampus can look more or less disorganized based on where along the A/P axis it is sampled.

Response 1:  Archival hippocampal paraffin embedded blocks were collected by different neuropathologists from Phoenix Children’s Hospital or Northwestern University Medical School, Chicago, IL prior to transfer to our laboratory. We stained tissue from each block with H&E and thionin revealing morphological characteristics consistent with the middle to posterior level of the hippocampus (see page 2, line 85, and Figure 1).

Point 2: Were DS and NTD hippocampal sections stained simultaneously with Thionin to control for variation in staining intensity between sets of slides?  Were slides all stained simultaneously, blind to experimental group? 

Response 2: All histo- and cytochemistry was performed at the same time using the same chemicals by an investigator blinded to case demographics, see lines 144 and 145.

Point 3: With no significant difference between numbers of neurons in dentate gyrus or CA1-3, what is the significance of an altered ratio of dentate to pyramidal cell numbers comparing NTD and DS brain? A similar observation was made in the SMI-32 and Doublecortin analyses, but the significance of these differences are not clear either.

Response 3: The significance of the altered ratios presented is to provide information about the degree of cellular marker expression/differentiation between regions in the hippocampal formation for each group.

Point 4: In the DCX tracings (Figure 8), the DS hippocampus looks dramatically different than the age-matched NTD hippocampi depicted, yet the numbers of neurons are not significantly different between groups. Do the authors think that there is a significant difference in the morphology of the hippocampus? Was this apparent in every DS brain compared to every NTD brain? Is there a way in which a morphologic abnormality can be quantified?

Response 4: We thank the reviewer for pointing out the miscommunication reported on lines 289- 292. We clarified this issue by restating the results as follows:

“At the light microscopic level, we observed fewer DG DCX-ir cells with increasing age in both groups (Figure 6E, F, and Figure S2A-F). After 61 weeks of age less DCX positive cells were observed in DS than in NTD (Figure 6E, F and Figure S2C, D) and none were visualized at either week 196 in DS or at 174 in NTD (Figures 6G, H and Figure S2E, F).  

Quantitative counts did not show a significant difference between the number of DCX-ir cells in DG, CA1 and CA2/3 fields between the DS and NTD groups (Figure 7; Mann-Whitney rank, p > 0.05). Counts also revealed that DCX positive cells in the DG were significantly greater compared to CA1 in the NTD group (Figure 7; Friedman repeated measures; p = 0.031), which was not seen in DS (Figure 7; Friedman repeated measures, p > 0.05).”

Point 5: The authors do not find Parv+ neurons in either DS or NTD cohorts. If they are absent in the NTD brains, why is their absence in the DS brain considered abnormal? With no evidence of a difference in Parv-expressing cells between DS and NTD brain, the logic for considering a role for these neurons in cognitive impairment in DS is not clear.

Response 5: We thank the reviewer for pointing out this oversight.  We corrected the statement on page 24, lines 536-539.

Point 6: Were calreticuin stains performed simultaneously under identical conditions between groups?  The NTD stains shown seem to have a lower background staining, suggesting the staining conditions are not identical.

Response 6: Sections from all cases (DS+NTD) were simultaneously immunostained for calretinin and for all other markers under identical conditions by an experimenter blinded to demographics to avoid batch-to-batch differences, as stated on page 4, lines 109 and 110. The difference in background staining is likely due to variations in length of fixation, premortem variables or overall tissue quality across cases.

Point 7:  What is the significance of a correlation between brain weight and DCX staining when it is also present in NTD brains?  It seems misleading to ignore that this correlation is present in both groups.

Response 7: We thank the reviewer for noting this inconsistency. We have corrected the text in the revised manuscript (see page 23, lines 472 and 473).

Point 8: Linear regression analysis comparing DG DCX cell numbers does not show more rapid loss of DCX neurons in DS brain compared to NTD brain (Figure 13B). Is there additional data supporting the difference stated in the discussion (page 29, lines 487-489)?  Why do the authors draw this conclusion?

Response 8: We thank the reviewer for pointing this discrepancy. We have corrected this statement in the revised manuscript, see page 23, lines 468-472 and page 26 lines 598-601.

Point 9: Are diffuse amyloid plaques really present in the neonatal NTD brain? Could this be a staining artifact?  Have diffuse amyloid plaques been described in the NTD brain before?  This is a surprising result if it is real.

Response 9: We apologize for any miss understanding of the protein composition of the plaque-like deposits reported in the postnatal hippocampus. Tissue was immunostained using the 6E10 antibody that recognizes both APP and Aβ and an antibody against Aβ1-42. We did not find any Aβ1-42 positive deposits in the hippocampus at any age in either the DS or NTD cases. These findings suggest that the plaque-like profiles observed contained APP, but not Aβ1-42. The latter is traditionally found in diffuse and senile plaques in older DS and AD cases.

Round 2

Reviewer 2 Report

The authors have addressed most of the issues that I raised in the initial evaluation of their manuscript; however, I still do not understand their interpretation of the presence of "diffuse APP plaques" in normal and DS infant brains. Is APP released from neurons to accumulate in diffuse plaques in the infant brain?  Do they hypothesize that these deposits represent secreted, cleaved fragments of APP?  Do these deposits disappear with age (they are not present in the brains of adults until late in life)?  Has this kind of staining been described in infants previously?  I am concerned that this represents some kind of artifact. 

See, for example, Davidson YS, Robinson A, Prasher VP, Mann DMA. The age of onset and evolution of Braak tangle stage and Thal amyloid pathology of Alzheimer's disease in individuals with Down syndrome. Acta Neuropathol Commun. 2018 Jul 4;6(1):56. In this paper, even diffuse amyloid does not appear in the DS brain until patients enter their teens. I appreciate that the authors of the paper I am referencing only stained for amyloid-beta, but I think the burden of proof is still on the authors of this manuscript to convince readers that the staining they demonstrate in DS and normal brains represents APP deposition in the brain parenchyma.  I still find it a surprising result and I am more surprised that it is not treated as such by the authors of this manuscript.

The authors state that no staining was observed when primary was omitted, but are they sure that the secondary that they used to perform the 6E10 staining was similarly tested?  Could the staining represent secondary antibody precipitate? Is this staining present throughout the brain or in other control tissues?  This is a very unusual result that cannot be included in this study without more significant exploration.  It may be safer to exclude the result if the authors are unable to perform additional characterization.
